# Entry of Phenuiviruses into Mammalian Host Cells

**DOI:** 10.3390/v13020299

**Published:** 2021-02-14

**Authors:** Jana Koch, Qilin Xin, Nicole D. Tischler, Pierre-Yves Lozach

**Affiliations:** 1CellNetworks—Cluster of Excellence and Center for Integrative Infectious Diseases Research (CIID), Department of Infectious Diseases, Virology, University Hospital Heidelberg, 69120 Heidelberg, Germany; jana.koch@med.uni-heidelberg.de; 2INRAE, EPHE, Viral Infections and Comparative Pathology (IVPC), UMR754-University Lyon, 69007 Lyon, France; qilin.xin@etu.univ-lyon1.fr; 3Fundación Ciencia & Vida, Molecular Virology Laboratory, Universidad San Sebastián, 7780272 Santiago, Chile; ntischler@cienciavida.org

**Keywords:** arbovirus, arthropod, bandavirus, cell entry, emerging virus, endocytosis, fusion, Heartland, phlebovirus, receptor, Rift, RNA virus, SFTSV, Toscana, Uukuniemi, uukuvirus

## Abstract

*Phenuiviridae* is a large family of arthropod-borne viruses with over 100 species worldwide. Several cause severe diseases in both humans and livestock. Global warming and the apparent geographical expansion of arthropod vectors are good reasons to seriously consider these viruses potential agents of emerging diseases. With an increasing frequency and number of epidemics, some phenuiviruses represent a global threat to public and veterinary health. This review focuses on the early stage of phenuivirus infection in mammalian host cells. We address current knowledge on each step of the cell entry process, from virus binding to penetration into the cytosol. Virus receptors, endocytosis, and fusion mechanisms are discussed in light of the most recent progress on the entry of banda-, phlebo-, and uukuviruses, which together constitute the three prominent genera in the *Phenuiviridae* family.

## 1. Introduction

*Phenuiviridae* in the *Bunyavirales* order is a large family of arthropod-borne RNA viruses that comprises 19 genera [1,2]. Phenuiviruses are unique in the sense that they infect a large spectrum of hosts, including humans and other vertebrates as well as invertebrates and plants. They usually spread to vertebrate hosts by blood-feeding arthropod vectors, such as sandflies and ticks and more rarely mosquitoes [3]. With over 100 identified members and a wide geographical distribution, phenuiviruses represent a global threat to human public health and livestock and agricultural productivity [4]. Many members cause serious diseases in humans and domestic animals. For instance, in severe cases, patients infected with Dabie virus (DABV) develop thrombocytopenia and hemorrhagic fever, resulting in a case-fatality rate of 10–30% [5]. Toscana virus (TOSV) can cause meningoencephalitis in humans, and Rift Valley fever virus (RVFV) acute hepatitis and fetal malformations in several mammalian hosts including cattle [6,7,8]. No vaccines or treatments against phenuiviruses are currently approved for human use.

Human activity, international trade, deforestation, and global warming are many of the factors favoring the spread of arthropod vectors to new regions as well as the viruses they carry. Many examples show that phenuiviral infections are no longer limited to tropical or developing countries. DABV, previously known as severe fever with thrombocytopenia syndrome virus (SFTSV), and Heartland virus (HRTV), are two closely related members in the *Bandavirus* genus transmitted by the *Haemaphysalis longicornis* and *Amblyomma americium* ticks, respectively [5,9,10]. DABV emerged in Henan and Hubei provinces, China, and HRTV in Missouri, USA, one decade ago [5,11,12,13,14]. The emergence of these two human pathogens led to a renewed interest in the study of Uukuniemi virus (UUKV), an uukuvirus originally isolated from the *Ixodes ricinus* tick in the village Uukuniemi, Finland, in the early 1960s [15]. UUKV is not associated with any disease in humans. It is a validated biosafety level (BSL)-2 surrogate that enabled major advances in many aspects of the cell life cycle of phenuiviruses with a higher biosafety classification, such as receptors, cell entry, and assembly [16,17,18].

The *Phlebovirus* genus includes members that also illustrate the need to seriously take phenuiviruses as potential agents of emerging and reemerging diseases. The phlebovirus Toscana was first isolated in 1971 from *Phlebotomus perniciosus* and *Phlebotomus perfiliewi* sandflies in the Tuscany region, Italy [19]. The virus is reemerging in the Mediterranean basin, as shown by the increasing number of outbreaks and reported cases in Spain, south of France, Italy, and Greece [6,20,21]. TOSV is currently the primary cause of arboviral diseases in humans in southern Europe during the summer [6,22]. Another example is RVFV, a phlebovirus transmitted essentially by *Aedes* and *Culex* mosquitoes [23]. The virus was discovered in the Great Rift Valley, Kenya, in 1930 [24] and has since spread across Africa and beyond in the 1970s to reach Madagascar and, more recently, Saudi Arabia and Turkey [7,8,25]. RVFV now presents the risk of introduction into southern Europe and Asia. It is listed as high-priority pathogens by the World Health Organization, for which there is an urgent need to develop diagnostics, therapies, and research [26]. The virus is in addition considered as a potential biological weapon by the US army. Overall, it is apparent that phleboviruses and other phenuiviruses are potential agents of emerging and reemerging diseases.

In 2016, we contributed to the first special issue on bunyaviruses from Viruses with a review on early bunyavirus-host cell interactions [27]. Since then, important discoveries have been made in this field. Taxonomic classification has evolved considerably to better reflect the variety of bunyavirus members, vectors, hosts, and diseases [1,2,28]. In this review, we therefore focus on the novel family *Phenuiviridae*. We address the most current knowledge and advances regarding the entry process of phenuiviruses, from virus binding to penetration into the cytosol. Most of the available information on how phenuiviruses enter cells comes from studies on only a few species, mainly banda-, phlebo-, and uukuviruses infecting animals (Table 1). Nothing is known about the penetration mechanisms in plant cells, and not much regarding penetration in arthropod cells. Hence, the discussion is limited to animal phenuiviruses and mammalian host cells. For information on phenuiviruses and their arthropod vectors, we recommend the following reviews [4,23,29,30,31].

## 2. Genomic and Structural Organization of Phenuiviral Particles

Phenuiviruses are enveloped by a lipid bilayer with a trisegmented single-stranded RNA genome, mainly of negative-sense polarity (Figure 1a) [4]. The viral RNA replicates exclusively in the cytosol and encodes at least four structural proteins [4]. The longest segment of genomic RNA, the L segment, encodes an RNA-dependent RNA polymerase, which is necessary to initiate virus replication after release of the viral genome into the cytosol. The medium segment, M, encodes a precursor polypeptide for the two envelope glycoproteins G_N_ and G_C_ (Figure 1b) [32,33]. Phenuiviruses also encode one or two nonstructural proteins, i.e., NSs and NSm. With the vector of transmission, the presence of an NSm protein appears as one of the main distinctions between tick- and dipteran-borne phenuiviruses [4]. Although phenuiviruses are quite distinct from each other, analysis of amino-acid sequences revealed that the structural proteins of phenuiviruses show a rather high similarity in general, up to 30–40%, compared to the nonstructural proteins, about 10–15% [4,5,34]. So far, none of the nonstructural proteins has been shown to be involved in virion cell entry and will therefore not be discussed here.

Proteolytic cleavage of the M precursor takes place in the endoplasmic reticulum or Golgi apparatus, where the virions assemble and acquire their lipid envelope. The exact location and mechanisms of G_N_ and G_C_ glycoprotein maturation and viral particle budding may differ between phenuiviral species and cell types. They often remain to be elucidated. The smallest segment S encodes the nucleoprotein N, which, together with polymerase and viral RNA, constitute the ribonucleoproteins found inside virions (Figure 1a) [4]. Phenuiviruses do not have a classical matrix or a rigid internal structure. The N protein therefore plays an important role in protecting the genetic information of these viruses. In recent years, the crystal structure of the N protein has been resolved for several phenuiviruses, providing new information on the mechanisms of ribonucleoprotein assembly [35].

On the surface of viral particles, the two envelope glycoproteins G_N_ and G_C_ are responsible for virion binding to the target cells and then acid-activated penetration into the cytosol [3]. Electron micrographs show that phenuiviral particles are globally spherical and heterogeneous in size, with a diameter ranging from 80 to 160 nm [4]. Electron cryo-tomography analyses of RVFV and UUKV revealed that the most regular particles have protrusions on their surface forming an icosahedral lattice with an atypical T = 12 triangulation (Figure 1c) [36,37,38].

**Figure 1 viruses-13-00299-f001:**
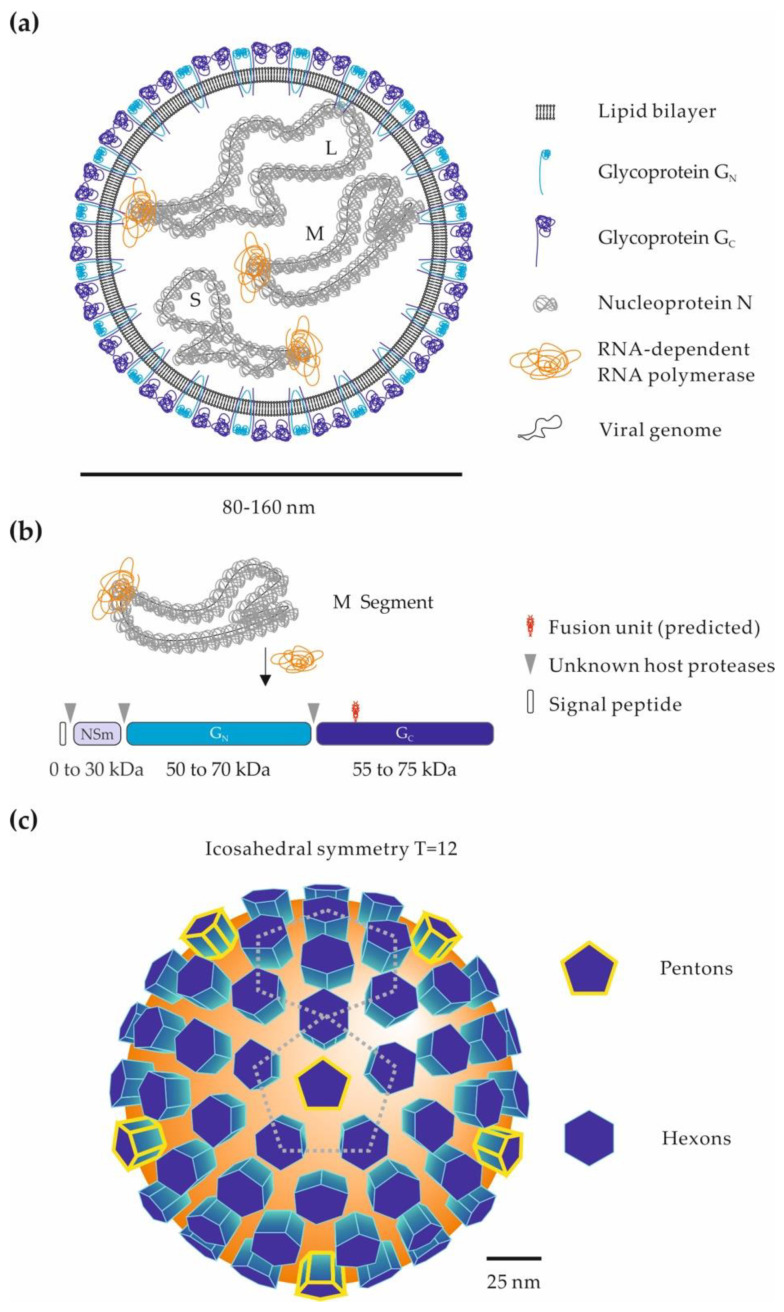
Phenuiviral particles and the glycoproteins G_N_ and G_C_. (**a**) Schematic representation of a phenuiviral particle. The three viral genomic RNA segments are named according to their size: S (small), M (medium), and L (large). (**b**) Proteolytic processing of the phenuivirus M polypeptide precursor. The M precursor and the glycoproteins G_N_ and G_C_ can vary greatly among phenuiviral species. In addition to G_N_ and G_C_, some phenuiviruses encode an additional nonstructural protein, NSm. Arrow heads indicate the cleavage sites by host cell proteases within the precursor. The position of the fusion peptide is given based on the crystal structure of the Rift Valley fever virus (RVFV) glycoprotein G_C_ [39]. (**c**) Arrangement of G_N_ and G_C_ glycoproteins on the surface of phenuiviral particles. Electron cryo-tomography analysis of RVFV [36,37] and Uukuniemi virus (UUKV) [38] viral particles shows an icosahedral lattice with an atypical T = 12 triangulation.

## 3. Cellular Receptors for Phenuiviruses in Mammalian Hosts

To initiate infection, viruses must first attach to target cells and then obtain access to the intracellular environment to replicate. The first step is highly dependent on the presence of surface receptors, such as proteins, lipids, or carbohydrates, to which viral particles bind [40]. Some of these receptors by themselves are capable of triggering the entry of viral particles into the cell. Others limit the free diffusion of virions and/or promote interactions with secondary surface molecules that are responsible for the entry of viral particles [40]. When viruses depend on numerous cellular surface factors for binding and entry, the primary receptor is often referred to as an attachment factor, and the secondary receptors are referred to as coreceptors. Only a few attachment factors and receptors are known for phenuiviruses, and very often, their role in cell entry remains to be discovered (Table 2).

Interactions between viruses and receptors are often specific and multivalent. Binding to several molecules of the same receptor, concentrating within microdomains, may increase the avidity of low-affinity interactions [40]. For example, glycoproteins and glycolipids present in the extracellular matrix of most mammalian cells are highly polar structures. Despite low affinity interactions, due to their electrostatic nature, these structures serve as attachment factors for many viruses, including phenuiviruses. Glycosaminoglycans (GAGs), such as heparan sulfates, have been shown to facilitate RVFV and TOSV infections (Table 2) [48,49,50]. Infection with these two viruses is greatly reduced in the presence of heparin, a competitor of GAGs on the cell surface. In addition, enzymatic digestion of heparan sulfate on the cell surface prior to exposure to these viruses produces similar results. Finally, cells deficient in heparan sulfate synthesis show a reduced sensitivity to RVFV [48,49]. Interestingly, the glycoproteins of an RVFV strain amplified in cell culture and used in one of these studies do not differ in basic amino acid composition from those on viruses isolated from infected animals [48]. This result suggests that the dependence of RVFV on heparan sulfates does not result from the adaptation of the virus to cell culture. However, the fact that some cell types remain permissive to infection even without GAGs indicates that phenuiviruses may use alternative receptors to enter cells.

A number of studies have indicated that many phenuiviruses can use the human C-type lectin DC-SIGN (also known as CD209) to target and infect dendritic cells (DCs) in the dermis (Table 2) [41,42,43,44,45]. In the presence of neutralizing antibodies, dermal DCs become resistant to infection with RVFV and UUKV [41]. The expression of DC-SIGN at the plasma membrane renders cells that are originally not permissive highly sensitive to several phleboviruses, including RVFV, UUKV, TOSV and Punta Toro virus [41]. The list of phleboviruses described to interact with DC-SIGN has since been extended to include bandaviruses. Recent studies demonstrate, among others, that lectin facilitates infection by DABV or rhabdoviral particles pseudotyped with DABV glycoproteins [42,43,44]. DC-SIGN represents an interesting molecular candidate for linking arthropod-derived viruses to the initial infection in the skin of the human host. This immune receptor is (i) mainly expressed on the surface of immature dermal DCs present in the anatomical site of transmission of these viruses and (ii) specialized in the capture of foreign antigens rich in mannose residues, such as the glycoproteins found in viruses produced from insects [3,41]. For these reasons, interactions between DC-SIGN and arthropod-borne pathogens are considered the most relevant, although several studies have suggested a role of lectin in infection by various microbes not propagated by arthropods [52].

Phenuiviruses generally possess numerous *N*-glycosylations on the particle surface, distributed between the envelope glycoproteins G_N_ and G_C_ [33,53]. For instance, RVFV has one glycosylation site in G_N_ and four in G_C_. RVFV appears to rely on the *N*-glycan sites N438 and N1077 in G_N_ and G_C_, respectively, for DC-SIGN-mediated infection [45]. It is tempting to draw a parallel with dengue virus in the *Flaviviridae* family and its envelope glycoprotein E. The engagement of multiple E molecules by homotetramers of DC-SIGN explains the high avidity of the interaction between the lectin and the viral particles [54]. The same is probably true for the interactions between DC-SIGN and phenuiviruses. In addition, the human C-type lectins L-SIGN and LSECtin, both closely related to DC-SIGN but expressed on the surface of the liver endothelium [55], are also used as receptors by several phenuiviruses, including RVFV, TOSV, UUKV, and DABV (Table 2) [42,43,46,47]. It is possible that L-SIGN and LSECtin, by acting as receptors on the surface of the liver endothelium, contribute to the hepatic tropism of some phenuiviruses.

Nonmuscle myosin heavy chain type IIA (NMMHC-IIA) has been proposed to act as an attachment factor for DABV (Table 2) [51]. This factor has been identified using a strategy combining coimmunoprecipitation and mass spectrometry analysis, using a fragment of the G_N_ ectodomain as bait. NMMHC-IIA usually has an intracellular localization but, in some cases, seems to be able to reach the outer surface of the plasma membrane, notably in human umbilical vein endothelial cells and Vero cells [51]. Silencing of the gene coding for NMMHC-IIA by small interfering RNAs (siRNAs) led to a significant decrease in infection by DABV. It is likely that NMMHC-IIA is not the only cellular receptor used by DABV. HeLa cells do not express this gene and are sensitive to infection [51]. However, ectopic expression of NMMHC-IIA resulted in an increased sensitivity of HeLa cells to DABV. It is still not known whether NMMHC-IIA serves as an entry receptor or simply as an attachment factor.

More recently, DABV has been observed in secreted vesicles exhibiting CD63, a marker of extracellular vesicles (Table 2) [56]. It appeared that virions within these vesicles are efficiently delivered to neighboring uninfected cells. This work was the first demonstration of the hijacking of the exocytosis machinery by a phenuivirus to ensure its transmission to surrounding cells in a receptor-independent fashion.

## 4. Internalization of Phenuiviruses into Cells

To be internalized and enter the host cell, phenuiviruses depend on the endocytic machinery. The number of studies on the endocytic receptors and pathways used by phenuiviruses has increased significantly in recent years (Table 3). However, after the attachment of virions to their receptor(s), the process of passage from the outside to the inside of the cell remains to be discovered for most phenuiviruses (Figure 2). By combining the use of fluorescently-labeled UUKV particles [57] and the expression of green fluorescent protein-tagged DC-SIGN molecules, it was possible to visualize virus-receptor interactions in living cells for the first time [41]. This model has made it possible to analyze the dynamics of these interactions and, in particular, to observe the recruitment of receptor molecules to the UUKV binding site on the cell surface [41]. This supports the hypothesis that some viruses aggregate their receptors at the site of contact, thus generating a receptor-enriched microdomain at the plasma membrane. Such a series of events is probably a precondition for local curvature of the plasma membrane and for the transduction of receptor-mediated signals to induce the internalization of virions into the endocytic machinery [40].

Cholesterol and other lipids also most certainly play a role in the early mechanisms of virus-receptor interactions by promoting the formation of anchoring sites for cellular cofactors. A study on DABV and HRTV entry into cells lacking glucosylceramide synthase activity suggests that this is also the case for phenuiviruses [58]. The authors show that the entry program of DABV, as well as that of HRTV, is greatly disrupted in the absence of this enzyme. Glucosylceramide synthase is involved in the maturation of ceramides, and the abrogation of its activity leads to a major change in the lipid composition of the plasma membrane and the resulting endosomal vesicles.

Amino acid motifs in the cytosolic tails of receptors generally define the identity of the endocytic pathway by which the cargo is internalized. These motifs serve as binding sites for specific proteins with functions in signaling and endocytosis [40]. The cytosolic tail of DC-SIGN contains several of these motifs, including two leucines (LL) critical for the internalization of the cargo by the lectin [59]. DC-SIGN mutants lacking the dileucine motif capture UUKV with the same efficiency as the wild-type receptor [41]. However, the virus is no longer internalized, and the infection is abolished. Interestingly, LL-based motifs are usually associated with adaptor proteins and clathrin-mediated uptake [60,61], and DC-SIGN was found to colocalize with microdomains enriched in clathrin [62,63]. Overall, this result shows that DC-SIGN serves in this case as an "endocytic" receptor, and not only as an attachment factor. Unlike the case for DC-SIGN, the endocytic function of L-SIGN is not required for UUKV infection [47]. Infection was similar whether cells expressed the wild-type lectin or its endocytic-defective mutant. This result indicates a fundamental difference in the use of DC-SIGN and L-SIGN by phenuiviruses for cell entry. In summary, DC-SIGN is an endocytic receptor for phenuiviruses whereas L-SIGN is an attachment factor. No other signal motif in receptors has been identified for phlebo-, uuku-, and bandaviruses with a function in virus internalization and productive entry.

**Table 3 viruses-13-00299-t003:** Cellular factors and processes important for phenuivirus infectious entry.

Virus	Fusion/Penetration	Cellular Factors in Viral Entry	
		Required	Not Required
DABV	pH~5.6 [64] 60 min (max) [64]	Actin [64], Ca^2+^ channels [65], clathrin [64], dynamin-2 [42,64], glucosylceramide synthase [58], LAMP1 [66], microtubules [64], serine proteases [42], Rab5 [64,66], Rab7 [64,66], SNX11 [66], vATPases [42]	Gangliosides series a and b [58], cathepsin B and L [42], caveolin-1 [64], cholesterol [64], lactosylceramide synthase [58], PAK1 [64], PI3K [42], Rab7 [58], Rac1 [64]
HRTV		Glucosylceramide synthase [58]	
RVFV	pH~5.7 [67] 16–24 min (t_1/2_) [67]	Actin [68], Ca^2+^ and K^+^ channels [68], caveolin-1 [69], cholesterol [69], clathrin [67], dynamin-2 [67,69], Na^+^/H^+^ exchangers [68], microtubules [68], PI3K [68], PKC [68], PLC [68], PP1/PP2A [69], RNASEK [70], vATPase [67,69]	Actin [67,69], cholesterol [67], clathrin [69], Na^+^/H^+^ exchangers [69], Eps15 [69], PAK1 [69], PI3K [69], Rac1 [69]
UUKV	pH~5.4 [71] 20–30 min (t_1/2_) [71]	BMP [72], clathrin [71], HDAC8 [73], LAMP1 [71], microtubules [71], PI3K [71], Rab5 [71], RNASEK [74], temperature [71], VAMP3 [74], vATPase [71]	Rab7 [71]

The cellular factors in blue have been described by independent studies as either required or not required in the cellular entry of the virus. BMP, bis(monoacylglycerol)phosphate; DABV, Dabie virus; HDAC, histone deacetylase 8, LAMP1, lysosome-associated membrane protein 1; PAK1, p21-activated kinase 1; PI3K, phosphoinositide 3-kinase; PLC, phospholipase C; PKC, protein kinase C; PP1/PP2A, protein phosphatase 1/2A; Rab5 and Rab7, Ras-related protein 5 and 7; RVFV, Rift Valley fever virus; RNASEK, ribonuclease K; TOSV, Toscana virus; UUKV, Uukuniemi virus; VAMP3, vesicle-associated membrane protein 3; vATPase, vacuolar-type proton pump ATPase.

A recent study suggests that DABV uses the clathrin-mediated endocytosis pathway to enter cells (Figure 2; Table 3) [64]. In cells expressing DC-SIGN, electron microscopy images show that UUKV particles enter clathrin-coated vesicles but not only these vesicles [41]. In non-lectin-expressing cells, such events are very rarely observed for UUKV [71]. Moreover, silencing of the clathrin heavy chain gene by siRNA has no significant impact on UUKV infection (Table 3) [71]. It is also unclear whether other phenuiviruses use the clathrin-mediated endocytosis to enter cells (Table 3). It has been proposed that a genetically engineered, nonspreading strain of RVFV relies on clathrin for productive entry [67]. Conversely, two reports suggested that the vaccine strain MP12 enters cells independently of clathrin. The first indicated a role of actin and many kinases with functions related to macropinocytosis (Table 3). These authors proposed that RVFV relies on macropinocytosis for entry [68]. The second study rather pointed out caveolin, implying that RVFV entry occurs through caveolin-dependent mechanisms (Table 3) [69].

**Figure 2 viruses-13-00299-f002:**
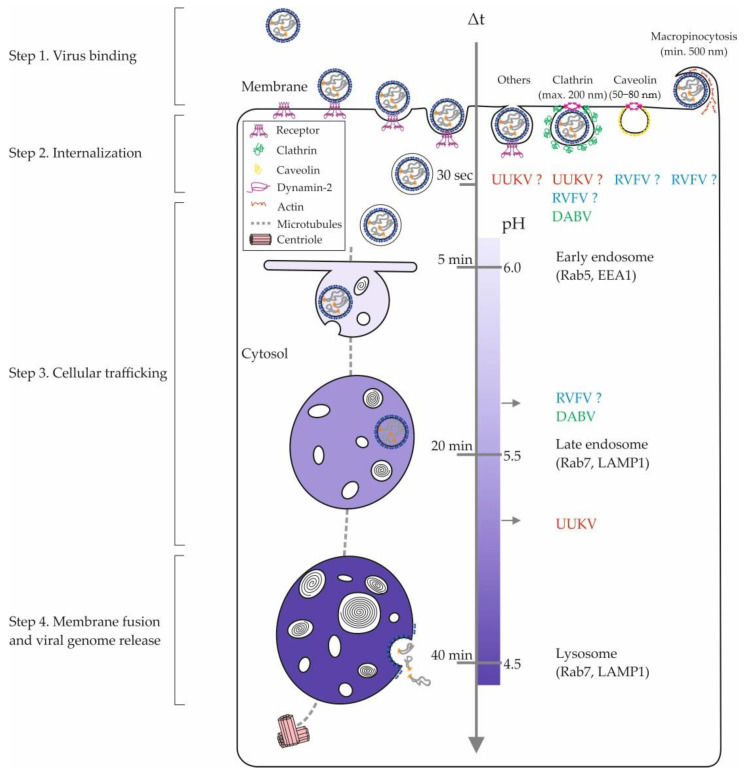
Phenuivirus endocytosis and intracellular trafficking. The internalization of phenuiviruses into mammalian cells involves diverse pinocytic pathways and hundreds of cellular factors. The figure shows an overview of the different cellular locations of phenuivirus penetration. The entry pathways of RVFV, UUKV, and DABV appear in blue, red, and green, respectively. Sizes refer to nonvirus cargo that are typically sorted into the clathrin-mediated endocytosis, caveolin-mediated pathway, and macropinocytosis [75]. In the middle, the scales indicate the time taken by a nonvirus cargo to go from the plasma membrane to an endosomal compartment and the corresponding endosomal pH values [75]. DABV, Dabie virus; EEA1, early endosome antigen 1; LAMP1, lysosome-associated membrane protein 1; Rab5 and Rab7, Ras-related protein 5 and 7; RVFV, Rift Valley fever virus; UUKV, Uukuniemi virus.

Beyond highlighting the amount of work that remains to better understand the internalization mechanisms of phenuiviruses, these studies most likely reveal the ability of these viruses to use alternative endocytic pathways in the same cell or in distinct tissues. This possibility is supported by a growing amount of data obtained for unrelated viruses, such as influenza virus. The variety of endocytic processes by which viral receptors are internalized into cells as well as the expression pattern of virus receptors on the cell surface certainly influence the ability of phenuiviruses to enter and infect cells and tissues through one or more endocytic pathways.

## 5. Intracellular Trafficking of Phenuiviral Particles

After binding to the cell surface and internalization into the cell, the phenuiviral particles are sorted into endosomal vesicles (Figure 2). The virions travel through the endocytic machinery until they reach the endosomal compartments from which they penetrate into the cytosol. Transport from early (EEs) to late (LEs) endosomes and then to lysosomes is a complex, highly dynamic biological process. It involves hundreds of cellular factors with a wide range of functions and is still far from fully understood [76]. It is accompanied by major changes in the protein and lipid composition, concomitant with acidification of the endosomal lumen from approximately pH 6.5 in EEs to 5.5–5.0 in LEs and lysosomes (Figure 2) [76]. Endosomal acidification has a central role in the penetration of a majority of viruses [77]. Numerous studies have demonstrated that phenuiviruses depend on low pH in intracellular vesicles for infection [42,67,69,71]. Many phenuiviruses are sensitive to the neutralization of endosomal pH by weak lysosomotropic bases, such as ammonium chloride and chloroquine, or by inhibitors of vacuolar-type proton pump ATPases (vATPase), such as bafilomycin A1 and concanamycin B, at extremely low concentrations (in the range of millimolar and nanomolar, respectively) (Table 3). Recent reports showed that cells deficient for RNASEK, a vATPase-associated factor, are insensitive to infection by RVFV and UUKV (Table 3) [70,74].

Several studies have clearly established that phenuiviruses transit through EEs during their journey in the endocytic machinery (Figure 2). The expression of dominant negative and constitutively active mutants of Rab5, a small GTPase necessary for EE trafficking and maturation, interrupts the intracellular transport of UUKV and therefore blocks infection by this virus [71]. Confocal microscopy images show the presence of UUKV and DABV particles in Rab5-positive EEs [64,71]. Overall, the data support the idea that phenuiviruses belong to a large group of viruses whose infectious entry depends on late endosomal maturation, known as late-penetrating viruses (L-PVs) [77]. Several phenuiviruses enter the cytosol by acid-activated penetration between 20 and 60 min after their internalization into cells (Table 3) [64,67,71], a period of time compatible with the maturation of LEs (Figure 2) [77].

The maturation of late endosomal vesicles occurs while trafficking along the microtubules from the periphery of the cell to the nucleus. The disruption of the microtubule network by inhibitors, such as nocodazole and colcemid, results in the blockade of productive infection by UUKV and DABV (Table 3) [64,71], indicating that these viruses require an intact microtubule network to infect host cells. Another indication that supports that phenuiviruses belong to the L-PV group is that the fusion of their envelope with the endosomal membrane requires a pH below 5.8–5.7 (Table 3) [64,67,71,78]. This acidity is typical of late endosomal vesicles [77].

It is thus apparent that many phenuiviruses penetrate their target cells from late endosomal compartments. However, the path of phenuivirus particles into the degrading branch of the endocytic machinery remains poorly characterized. VAMP3, a v-SNARE fusion protein associated in part with late endosomal vesicles [79], has been shown to be essential in the late intracellular trafficking of UUKV (Table 3) [74]. UUKV and DABV particles were visualized by confocal microscopy in intracellular vesicles positive for Rab7 (Table 3) [64,71], which is considered the most critical small GTPase for LE trafficking and maturation [80]. While expression of the dominant negative T22N mutant of Rab7 has no effect on the infectious entry of UUKV, the constitutively active mutant appears to facilitate infection [71]. The presence of UUKV and DABV in endolysosomes [41,64,71] raises questions about the path taken by these viruses to reach these late compartments (Table 3). The formation of endolysosomal vesicles results, at least in part, from the maturation of LEs under the control of Rab7. Several reasons could explain the inefficiency of mutants in blocking endogenous Rab7 activity, such as their mislocalization in the cell or the existence of several Rab7 isoforms. It is tempting to postulate that some phenuiviruses may use alternative, undescribed pathways from the late endocytic machinery to reach their compartments from where they enter the cytosol.

## 6. Fusion and Penetration of Phenuiviruses into the Cytosol

The viruses that enter cells by endocytosis must ultimately cross the membrane of endosomes to release their genome and accessory proteins into the cytosol. Phenuiviruses achieve this step with the viral glycoprotein G_C_, which mediates the fusion of their envelope with the endosomal membrane [53]. Acidification is sufficient to trigger the fusion of RVFV and UUKV in cell-free in vitro experiments [67,72]. It has been shown that exposure of RVFV to acidic buffers in the absence of any target membrane leads to conformational and oligomerization changes in G_C_ [67]. In this respect, endosomal acidification serves as a major cue to trigger fusion of many enveloped viruses but, by itself, can be insufficient. Among other factors, such as nonproton ions, intracellular receptors, specific lipids in the target endosomal membranes, or proteolytic cleavage of viral envelope proteins are sometimes required [76]. It seems that in addition to acidification, the infectious penetration of DABV relies on calcium and the cleavage of the glycoproteins G_N_ and G_C_ by serine proteases (Table 3) [42,65]. However, proteolytic processing may be a specificity of DABV as no other phenuivirus has been shown to rely on the cleavage of G_N_ and G_C_ for infection. Additionally, glycerolphospholipids and bis(monoacylglycerol)phosphate, a constituent lipid of specific LE subpopulations, facilitate RVFV and UUKV fusion with liposomes, respectively [72,81].

There are at least three distinct classes of viral proteins with the capacity to mediate membrane fusion (classes I to III), each with its own mechanistic and structural specificities [82]. Significant progress has been made in recent years in the structural characterization of the G_C_ fusion glycoprotein of phenuiviruses. The crystallographic structure of the ectodomain of the RVFV G_C_ protein has been obtained in its two states, pre- and postfusion, with a resolution of less than 2.0 Å [39,81]. The structure of the postfusion form of G_C_ was also determined for DABV and HRTV [83,84]. All these data indicate that the G_C_ viral glycoprotein of phenuiviruses belong to the group of class II membrane fusion proteins [32].

Recently, the ectodomain of the RVFV and DABV glycoprotein G_N_ has been crystallized and analyzed by X-ray crystallography [53,85], revealing a three-domain fold reminiscent to that of the alphavirus E2 and hantavirus G_N_ glycoprotein [86,87]. This important information made it possible to fit the crystalline structures of RVFV G_N_ and G_C_ in tomographic reconstructions of viral particles when inserting into target membranes after acidification [53]. A fusion model was proposed based on computer simulations predicting the passage from one structure to another. When the virus is in neutral pH medium, G_N_ protects the hydrophobic fusion loops from the G_C_ glycoprotein. G_N_ would thus have a protective role, preventing premature fusion of virions during passage through the Golgi network and virus egress. Once the viral particles are in the endosomal lumen, close to a target membrane, and at a pH below 5.7, G_N_ is believed to reorient to the sides. G_C_ is then released and extends to the outer leaflet of the target endosomal membrane to insert its fusogenic unit. This model is very similar to the one proposed for the fusion of flaviviruses, which are unrelated enveloped viruses with a class II viral fusion glycoprotein [82]. Interestingly, antibodies exhibit much higher neutralizing activities in vitro and in vivo when they target G_N_ rather than G_C_ [88,89,90]. It has been proposed that anti-G_N_ neutralizing antibodies block the fusogenic rearrangement of the G_N_-G_C_ array by targeting the G_N_ distal domain [91,92].

The ectodomain of class II fusion proteins has three subdomains (I to III) and is connected to a transmembrane cytosolic tail by a short stem [93]. Analogous peptides from domain III and the stem have been successfully used to block infection by flaviviruses and by other class II viruses, such as alphaviruses and hantaviruses [94,95,96]. Such peptides have been shown to interfere with intramolecular interactions in viral fusion proteins during conformational rearrangements leading to membrane fusion. The presence of these peptides probably maintains the fusion protein in an intermediate conformation, preceding the postfusion stage, and thus prevents membrane fusion and viral infection. Similar strategies have been employed with some phenuiviruses and have given identical results. For example, peptides targeting the stem of the G_C_ glycoprotein block RVFV infection [97].

Histidines in fusion proteins, including class II proteins of viral origin, act as sensors of acidification in the endosomal lumen. These residues often define the optimal pH of viral fusion [98]. The local environment of these histidines influences their pKa, with values ranging widely from 4.5–7.3, and ultimately the optimal pH for viral particle fusion [99]. Such histidine residues have been identified in the RVFV G_C_ glycoprotein by mutational analysis [67]. The pH threshold for penetration has been determined for several phenuiviruses using approaches that measure the fusion between viruses and liposomes, viruses and cells, or between cells and cells when they express the G_N_ and G_C_ glycoproteins on their surface. Fusion typically occurs at pH 5.4 for UUKV, 5.6 for DABV, and 5.7 for RVFV [64,67,71,72]. Ultimately, the main factor triggering fusion of most phenuiviruses appears to be the endosomal pH.

In sum, fusion of the viral and endosomal membranes is a closely coordinated mechanism in time and space [82]. During this process, viral fusion proteins undergo multiple conformational changes. They target and harpoon the lipid bilayer of the endosome via their fusion subunit. Gradually, they pull the membranes of the endosome and virion toward each other in successive steps of narrow apposition, hemifusion, and fusion [76,82,93]. This results in the opening of pores in the endosomal membrane through which viral material is released into the cytosol. The cell is then infected, and viral replication begins.

## 7. Concluding Remarks and Future Perspectives

In this review, we summarized and provided an update of the current knowledge on the early interactions between phenuiviruses and their target cells, from the attachment of viral particles at the cell surface to their fusion and penetration into the cytosol. Although each member most likely has distinct specificities and needs, it appears that many phenuiviruses have in common the dependence on late endosomal maturation and low pH for the infection of target cells. However, it is clear that many aspects of phenuivirus penetration remain to be elucidated. High-throughput screens aiming to selectively inactivate genes of the human genome, for example, with approaches combining haploid cells and RNAi or the CRISPR/Cas9 system, should help to identify new factors and cellular processes important for phenuivirus entry. Such approaches have started to be applied to RVFV and UUKV [49,74,100].

To target and infect a large number of different tissues and species, phenuiviruses can use multiple receptors. A few have been found in humans and other vertebrates but none in arthropods. A detailed analysis of the receptors of these viruses in both vertebrate and arthropod hosts, is therefore essential to better understand the underlying infection processes. Consequently, only the combined use of new in vitro models with ex vivo and in vivo approaches will allow us to improve our knowledge of the transmission, entry, and spread of phenuiviruses.

The characterization of viral particles transmitted by arthropods to mammals is also an important objective. Cell biology in arthropods differs significantly from that in mammals. The lipid composition of the viral envelope, and the nature of oligosaccharides in the glycoproteins on the surface of virions are factors that influence the identity of target cells, interactions with receptors, endocytosis mechanisms, and membrane fusion. For example, it has been shown that the glycosylation and conformation of the G_N_ glycoprotein of UUKV differ greatly depending on whether the virus is produced in tick or mammalian cells [16]. In addition, tick cell-derived UUKV has a higher infectivity than its counterpart amplified in mammalian cells [16].

Studies on phenuiviruses and on other arthropod-borne viruses often involve mammalian cell-derived virus stocks, which are less relevant for studying the transmission and initial infection of these pathogens.

Ideally, preventing the spread of phenuiviruses requires approaches that target the early stages of infection. While improving our knowledge of the host spectrum and tissue tropism is essential, at the molecular level single inhibitors cannot accurately define a cellular pathway. Perturbants very often have many side effects or simply alter different processes in the cell. Only a combination of inhibitors with a well-defined profile used in combination with qualitative and quantitative approaches to visualize and analyze the very first instants of infection will make deciphering the entry mechanisms of phenuiviruses possible. These are the keys to improving our understanding of the spread of these viruses and ultimately developing new antiviral strategies.

## Figures and Tables

**Table 1 viruses-13-00299-t001:** Classification within the family of *Phenuiviridae* [1,2].

Genus	Species	Representative Species
***Bandavirus***	7	Dabie bandavirus [previously named severe fever with thrombocytopenia syndrome virus (SFTSV)], Heartland bandavirus (HRTV)
*Beidivirus*	1	Dipteran beidivirus
*Cugovirus*	2	Citrus coguvirus
*Entovirus*	1	Entoleuca entovirus
*Goukovirus*	3	Gouleako goukovirus
*Horwuvirus*	1	Horsefly horwuvirus
*Hudivirus*	1	Dipteran hudivirus
*Hudovirus*	1	Lepidopteran hudovirus
*Ixovirus*	3	Blackleg ixovirus
*Laulavirus*	1	Laurel Lake laulavirus
*Lentinuvirus*	1	Lentinula lentinuvirus
*Mobuvirus*	1	Mothra mobuvirus
*Phasivirus*	5	Badu phasivirus
***Phlebovirus***	60	Rift Valley fever phlebovirus (RVFV), Punta Toro phlebovirus (PTV), Sandfly fever Sicilian phlebovirus (SFSV), Sandfly fever Naples phlebovirus (SFNV), Toscana phlebovirus (TOSV)
*Pidchovirus*	1	Pidgey pidchovirus
*Rubodvirus*	2	Apple rubodvirus 1
*Tenuivirus*	8	Rice stripe tenuivirus
***Uukuvirus***	17	Uukuniemi uukuvirus (UUKV)
*Wenrivirus*	1	Shrimp wenrivirus

The genera and species highlighted in this review appear in bold and underlined, respectively.

**Table 2 viruses-13-00299-t002:** Receptors for phenuiviruses in mammalian hosts.

Receptor/Cofactor	Species	References
DC-SIGN	DABV, ppDABV, PTV, RVFV, TOSV, UUKV	[41,42,43,44,45]
L-SIGN	ppDABV, RVFV, TOSV, UUKV	[42,43,46,47]
LSECtin	ppDABV	[43]
Heparan sulfates	RVFV, TOSV	[48,49,50]
NMMHC-IIA	DABV	[51]

DABV, Dabie virus; DC-SIGN, dendritic cell-specific intercellular adhesion molecule-3-grabbing non-integrin; L-SIGN, liver-specific intercellular adhesion molecule-3-grabbing non-integrin; LSECtin, liver and lymph node sinusoidal endothelial cell C-type lectin; NMMHC-IIA, nonmuscle myosin heavy chain IIA; ppDABV, rhabdovirus pseudotyped with the glycoproteins G_N_ and G_C_ of Dabie virus; PTV, Punta Toro virus; RVFV, Rift Valley fever virus; TOSV, Toscana virus; UUKV, Uukuniemi virus.

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
