# Peer review of "Entry of Phenuiviruses into Mammalian Host Cells"

_viruses, 2021, doi:10.3390/v13020299_

Round 1

Reviewer 1 Report

Title: Entry of Phenuiviruses into Mammalian Host Cells

In this manuscript, Koch et al., described a comprehensive perspective of cell entry by Phenuiviruses. The authors organized massive information very well by separating the cell entry steps into the four steps. This excellent manuscript will be great help of many readers including me. I can point out only one small suggestion due to completeness.

Minor point

  1. The author mentioned about UUKV produced from tick cells with reference 25 in the final section. Is this all information about arthropod vector-derived Phenuiviruses? Your manuscript title definitely includes “mammalian cells”, but Phenuiviruses is arboviruses transmitted by the vectors. Could the author add a little bit more information about the vectors or introduce other review articles about them?

Reviewer 2 Report

Koch et al have reviewed entry processes of a selection of phenuiviruses. They describe the current state of literature on RFVF, UUKV and DABV regarding viral entry in mammalian cells. This review is well-written and provides a nice overview.

Major comments

  1. Line 34-40: This section is a bit vague with the authors using terms like ‘many members’, ‘some phenuiviruses’ and ‘certain’. It would be helpful for the reader if the authors gave examples of specific viruses. E.g. please specify which viruses cause serious disease in human/animals, which viruses have caused these outbreaks, which are classified as biological weapons etc.
  2. In my opinion, the introduction could provide a bit more epidemiological background for the viruses highlighted in the rest of the manuscript. Especially UUKV is missing in the introduction section.
  3. It would be helpful for the reader to get a sense of the genetic diversity between various phenuiviruses, and specifically for the discussed members.
  4. Table 1: The header should specify that this is not the whole Phenuiviridae family. It would also be helpful to highlight the viruses that are discussed in this manuscript (e.g. by underlining them)

Minor comments:

  1. Line 13 and other places: I am not sure that ‘isolates’ is the correct term here. Generally, that term is reserved from virus that is specifically recovered from patients or infected animals/vectors. In other places ‘species’ is used, which seems a more appropriate term.
  2. Line 16: “arthropod reservoir” may not always be accurate. Suggest to use “arthropod vectors” instead
  3. Line 17: consider ‘some’ phenuiviruses represent a global threat
  4. Line 145: “A number of studies have indicated that many phenuiviruses can use the human C-type lectin DC-SIGN (also known as CD209) to target and infect dendritic cells (DCs) in the dermis [27].” – I’d expect more than 1 reference after such a sentence
  5. Line 279: “Endosomal acidification has a central role in the activation of a majority of viruses” – ‘activation’ isn’t quite the right term here. The low pH induces a conformational change required for membrane fusion, so it’s required for entry/infection, but I wouldn’t say for ‘activation’.

Reviewer 3 Report

Summary

Koch et al. review the entry of phenuiviruses. Phenuiviruses belong to the bunyavirales and are arthropod borne. Due to habitat changes and various human interferences emerging phenuiviruses are expected to pose an increasing threat. The review first introduces the organization of the phenuivirus genome and proteome and the structural organization of the particles. The authors then cover virus binding to receptors, virus internalization by endocytosis, the trafficking of the virus through the endo-lyosomal system and fusion between virus and endosomal membranes. Finally, the authors discuss open questions in phenuivirus entry and also highlight some limitations of current approaches. The topic of phenuivirus entry is important and the review is well structured. However, the authors miss the opportunity to carefully review the literature and to extract and discuss in detail the host factors that are facilitating phenuivirus entry. Additionally, some of the figure and tables require major improvements.

Major

The step of internalization is arguably critical for virus entry. As the authors are rightfully pointing out, viruses have been shown to engage with different endocytic processes and recruit different sets of cellular machineries to overcome the plasma membrane. In case a virus can bind to multiple different receptors, the internalization process may also depend on the receptors and co-receptors that the virus is engaging with per particular host cell. These variables all complicate the study of virus entry mechanisms. To approach the question, the cellular pathways and components are perturbed and virus entry or a downstream readout of it is analyzed in comparison to control cells. In a review on virus entry, I would expect a detailed discussion of the literature on how various perturbations affect virus endocytosis. Unfortunately, in this review some of this most important information is only mentioned in table 3 but not discussed in the text. Moreover, it would be useful for the reader to see more of these cellular components that are necessary for infection incorporated into figure 2. Based on table 3 there appear to be significant differences between various viruses. It should be discussed why these viruses display preferences towards e.g dynamin. Table 3 lists a series of signaling molecules, unfortunately the role they play in virus infection is not further discussed. The following paragraph on endosomal sorting includes somewhat more detail and is more useful.

More generally, after reading this review it is not clear to me how recent discoveries clarified the molecular mechanisms that contribute to phenuivirus entry. The authors fail to extract common patterns and molecular players from the literature that are necessary for the early steps of phenuivirus infection. This review would benefit from a more systematic review of the literature. Apart from the requirement for an acidic environment in a late endosomal compartment for virus fusion there appears to be little consensus and molecular understanding of how phenuiviruses enter cells. The phenuivirus family is certainly understudied but a couple large scale screens have been performed in the past and a wealth of data ought to be available. Despite the importance of the screens, they are barely covered in this review.

The figure 2 is central to this review since it summarizes the current understanding of the entry pathways that different phenuiviruses take. It should really aim at providing a summary of the current mechanistic understanding. Unfortunately, the figure is not different form similar figures in earlier review articles. Along the same lines the table 3 list cellular components that are needed by different phenuiviruses but references are missing or not clearly linked to each molecule. Nor are the components explained in the text or included in the overview figure. Both figure and table need to be significantly improved.

Specific comments

Table 3: this table needs to be expanded and improved. E.g. for RVFV MP12 strain a dominant negative version of Eps15 was overexpressed to block clathrin mediated endocytosis (Harmon et al. 2012, JVi) and the authors list Eps15 as not required. From these results I would rather conclude that clathrin mediated endocytosis in general is not required. At the same time clathrin and dynamin are listed as required but the reference is missing. In the corresponding section of the text the reference is absent too. This is confusing. Along the same line, actin is mentioned multiple times in the table but it is not covered in the text or referenced. Actin turnover is important for the entry of a large number of viruses and should be discussed.

Figure 2: the panel suggests that viruses above the 200nm size limit are not internalized via clathrin coated pits. For VSV it has been shown that even larger particles can recruit the clathrin machinery but become more dependent on actin polymerization for internalization (Cureton et al. 2010 PLoS Pathogens). Also, Dynamin is shown in this figure, but the potential role of dynamin in phenuivirus endocytosis is not discussed in the text. Moreover, there is a time axis in the middle of figure 2 and timestamps are indicated. It is not clear whether the first time stamp for receptor binding should be 15” (for seconds) or 15’ (for minutes). Steps further down are 30’ for endocytosis 5’ for early endosomes. This is confusing. The figure should be updated and the legend should state whether it is seconds of minutes. For DABV the time required for entry is listed as >60min in table 3, while in the figure DABV is emerging from endosomes at a position faster than the 20 minutes time stamp. These conflicts should be resolved.

Line 239: it is a really interesting observation that the di-leucine mutant DC-SIGN still binds UUKV but does not support the internalization of the virus. The authors, however, miss to explain why the di-leucine motif is important and how it links to AP-2 and the clathrin machinery. It should be discussed to what extend these results suggest that clathrin mediated endocytosis is a valid entry path for phenuiviruses. This type of experimental evidence should be collected for each virus to give a more detailed description of endocytic process they use.

Line 253: this sentence is misleading. Vesicles that are coated with clathrin are typically referred to as clathrin coated vesicles. Endosomes are typically bigger and composed of multiple domains. They can engage with clathrin coats for fusion and fission but are not considered fully coated.

Line 255: the sentence is not clear. The siRNA does reduce the mRNA expression but does not inactivate the gene.

Paragraph 262-269: needs to be rewritten. The plasticity in virus uptake mechanisms can certainly be acknowledged. It has been observed for a other viruses before. But rather than suggesting that anything is possible, this review should be focusing on the available molecular evidence for and against certain pathways and discuss why the virus-host interaction evolved that way.

Line 272: It is not clear what the authors mean by “circulating through endosomes”. But it suggests that viruses randomly travers various compartments until they happen to end up in late endosomes from where they escape to the cytosol. This is in conflict with the models of endosome maturation and endosomal sorting that follow molecular principles rather than chance as the authors are pointing out. The sentence should be changed.

Line 333: The text states that for DABV endosomal cathepsin activity is needed for processing of viral glycoproteins. But in table 3 cathepsins are listed as not required. They cite two papers. Hoffman et al. found that cathepsin inhibitors did not block DABV infection while a serine protease inhibitor did inhibit infection. Li et al found that calcium channel blockers reduced virus infection, replication and death in hospitalized patients, but does not provide data on cathepsins. This potentially contradictory data should be discussed. It is also not clear whether there is actually evidence that proteolytic processing of fusion proteins in the late endosome is needed for some phenuiviruses.

Line 386: is not clear how the authors conclude that “it is evident that hundreds of cellular factors with a wide range of biological functions are involved in the process of phenuivirus infectious entry”. Here they review a couple factors in detail but it is not clear which factors are essential for virus entry.  But it they do not provide evidence that it is hundreds of host factors.

Line 405: They mention the genome wide screening approaches to identify host factors involved in phenuivirus infection. They are citing one study where it was found that glycosaminoglycans are important for Rift Valley Fever Virus binding, but again in table 3 or in figure 2 this data is not included. The authors own siRNA screen on UUKV identified VAMP3 as important host factor for UUKV penetration from late endosomes, but decide to not discuss the role of VAMP3 in this review. The authors should more thoroughly screen the literature, condense the information and present a consistent summary of the cellular processes that intersect with virus infection.

Line 427: the authors lament that many studies are plagued from incomplete data, particularly the use of chemical inhibitor without validation of results by orthogonal approaches, like gene knockdown or knockout or overexpression of dominant negative constructs. These observations may be justified to some extent. But this review misses in large parts to label the insights that were obtained from inhibitor studies that the authors think were premature or flawed.

Round 2

Reviewer 3 Report

The authors addressed my comments, but some issues remain.

Major

Line 429: “It is evident that hundreds of cellular factors with a wide range of biological functions are involved in the process of phenuivirus infectious entry.” From reading this review it is not evident that hundreds of cellular factors are involved. With all due respect, table 3 lists around three dozen factors that are established to be linked to infection. And for the factors labeled in blue in this table it isn’t even clear whether they are needed or not. So, it is NOT evident why hundreds of additional factors would be needed for infection. Viruses could have evolved to be dependent on a minimal set of cellular machineries for entry. One could argue that the fact that despite extensive screening and research only few factors have been established so far, suggests that there will unlikely be hundreds more. Of course, the authors are free to speculate that hundreds of additional factors might be involved. But they should not pretend that they provided evidence to suggest that there will be hundreds of factors necessary for infection.

Specific comments

Line 195: the sentences should be rearranged. The authors note that some results were obtained from studies with rabdoviruses pseudotyped with Gn and Gc. Are they suggesting that the results are not valid or expected to be different if wt DABV was used? If so, they should rationalize why they believe the studies with pseudotyped virus could me misleading. Alternatively, the sentence should be deleted.

Line 265: this sentence is misleading and should be adjusted. It is correct that biochemical evidence for a direct interaction between DC-SIGN and AP-2 via the di-leucine motif has not been reported. There is, however, accumulating evidence that DC-SIGN di-leucine motif is important for clathrin mediated endocytosis (CME). The surface expression of the LL/AA mutant was increased, indicative of reduced endocytosis (PMID 11825573). The uptake of DC-SIGN cargos like carbohydrate-binding domain antibodies (PMID 11859097, 19585517), HIV-1 gp120 coated quantum dots (PMID 17388641), were reduced with the LL/AA mutant and also reduced when CME was targeted by siRNA. Moreover, there is a body of research by Ken Jacobson confirming the colocalization of DC-SIGN microdomains with clathrin (e.g PMID 21641311, 28128492).

Line 328: the term “chemical molecules” is a pleonasm, it should read something like “small molecule inhibitors” or “inhibitors”.

Line 392: the sentence is incomplete and should be rearranged, e.g. by ending with “when they target Gn rather than Gc.”
